# Dynamic molecular oxygen production in cometary comae

Yunxi Yao[1] & Konstantinos P. Giapis[1]

Abundant molecular oxygen was discovered in the coma of comet 67P/Churyumov–Gerasimenko. Its origin was ascribed to primordial gaseous $O_2$ incorporated into the nucleus during the comet's formation. This thesis was put forward after discounting several $O_2$ production mechanisms in comets, including photolysis and radiolysis of water, solar wind–surface interactions and gas-phase collisions. Here we report an original Eley–Rideal reaction mechanism, which permits direct $O_2$ formation in single collisions of energetic water ions with oxidized cometary surface analogues. The reaction proceeds by $H_2O^+$ abstracting a surface O-atom, then forming an excited precursor state, which dissociates to produce $O_2^-$. Subsequent photo-detachment leads to molecular $O_2$, whose presence in the coma may thus be linked directly to water molecules and their interaction with the solar wind. This abiotic $O_2$ production mechanism is consistent with reported trends in the 67P coma and raises awareness of the role of energetic negative ions in comets.

[1] Division of Chemistry and Chemical Engineering, California Institute of Technology, Pasadena, California 91125, USA. Correspondence and requests for materials should be addressed to K.P.G. (email: giapis@cheme.caltech.edu).

Although oxygen is the third most abundant element in the universe, its molecular form (dioxygen, $O_2$) is very rare. Molecular oxygen has only been detected in two interstellar clouds, the Orion Nebula[1] and the $\rho$ Oph A dense core[2]. In contrast to Earth, where oxygenic photosynthesis has made $O_2$ abundant, only tenuous amounts of dioxygen are found elsewhere in our solar system, for example, in the moons of Jupiter[3], Saturn[4] and on Mars[5]. Remarkably, molecular oxygen was detected recently in the coma of comet 67P/Churyumov–Gerasimenko by the Rosetta spacecraft, with local $O_2$ abundances ranging between 1 and 10% relative to water[6]. The $O_2$ was proposed to be of primordial origin in conflict with Solar System formation theories[6].

Understanding the origin of molecular oxygen in space is important for the evolution of the Universe and the origin of life on Earth[7–9]. In fact, molecular oxygen in abundance has been suggested as a promising biomarker[10]. Interstellar and cometary oxygen is strongly bound chemically to other elements in compounds, such as $H_2O$, $CO_2$, CO, silicates and metal oxides. Release of $O_2$ from these reservoirs is practically difficult and energetically very expensive. Energetic particles, such as photons, electrons and ions, exist in abundance in astrophysical environments and can initiate dissociation reactions that ultimately produce $O_2$. Apropos, photolysis and radiolysis of water, solar wind interactions with the nucleus surface, and gas-phase collisions in the coma have been considered but found deficient in explaining the origin of $O_2$ in the coma of comet 67P (ref. 6). Despite their cometary abundance, energetic molecular ions such as $H_2O^+$ and $H_3O^+$ have not been discussed in the context of $O_2$ production.

When energetic molecules collide with surfaces, they may dissociate promptly, undergo electronic excitation or participate in surface reactions. The latter include Eley–Rideal (ER) reactions, where energetic projectiles collide with surfaces and react with adsorbates to produce projectile-adsorbate molecules without equilibration with the surface[11]. This dynamic process is driven by the projectile energy, a large fraction of which is carried away by the product molecule. Despite the implied similarity, the ER reaction process is different from sputtering (physical ejection of surface matter): a targeted new bond is formed, typically at hyperthermal incidence energies (10–200 eV). Notably, ER reactions have no surface temperature dependence and thus could be important in cometary environments during active periods, when energetic molecular ions are generated through interactions with the solar wind. Indeed, 'accelerated' water ions have been discovered in the inner coma of comet 67P (refs 12,13). These ions possess kinetic energy between 120 and 800 eV, and impact and sputter the nucleus surface at fluxes comparable to the typical solar wind flux[13]. The outer crust of the 67P nucleus facing the Sun is dehydrated[14], thus exposing mineral surfaces to the ions.

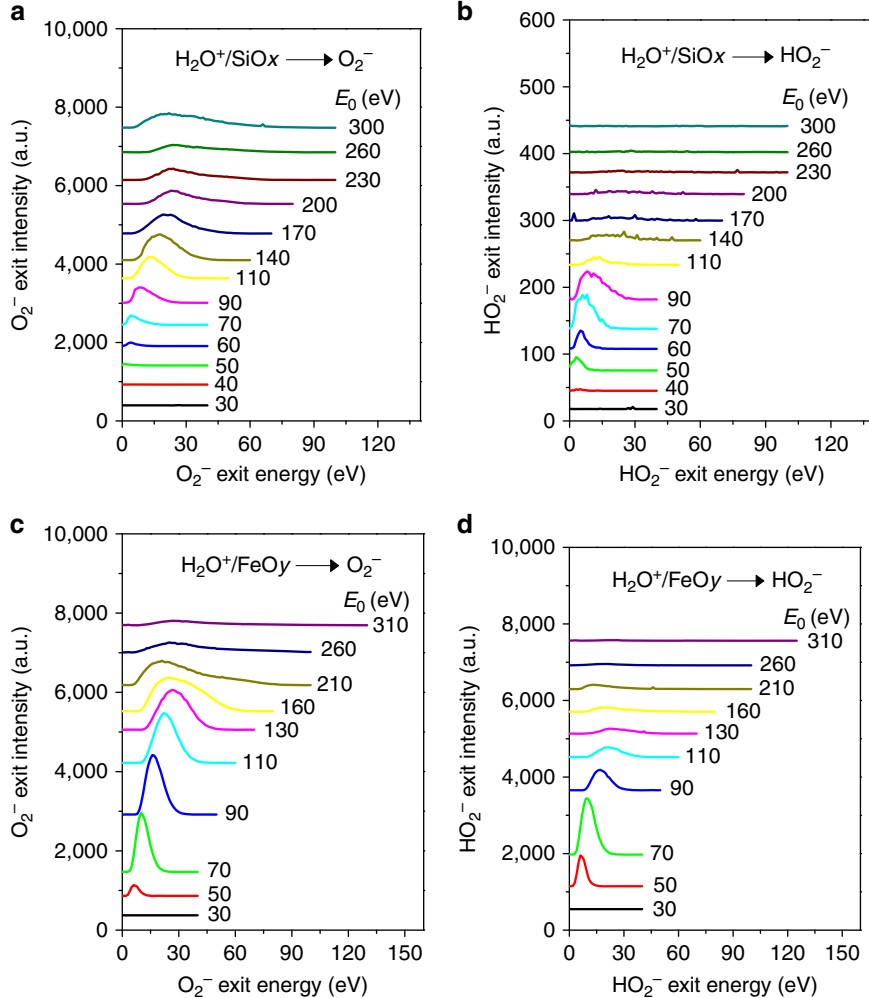

**Figure 1 | Production of $O_2^-$ and $HO_2^-$ from energetic $H_2O^+$ bombardment of oxides.** Energy distributions of (**a,c**) $O_2^-$ and (**b,d**) $HO_2^-$ scattered from native Si oxide (**a,b**) and Fe oxide (**c,d**) following bombardment by $H_2O^+$ at various incidence energies. Physical sputtering contributions to the $O_2^-$ signal become visible at high energies ($E_0 > 250$ eV for $SiO_x$, $E_0 > 150$ eV for $FeO_y$), when the dynamic $O_2^-$ peak dies out.

Minerals found on the comet, such as olivine and pyroxene silicates[15,16] and Fe/Ni oxides[17], are oxidized offering another potential source of oxygen. Thus, collisions of energetic water ions with oxidized minerals on the nucleus surface are probable, with several possible outcomes: (1) collision-induced dissociation (CID) of $H_2O^+$ produces atomic O, atomic H and OH radicals and ions; (2) physical sputtering ejects mineral constituents, including metal and oxygen atoms; and (3) collisional excitation of $H_2O^+$ drives an intramolecular water-splitting reaction[18], producing molecular $H_2$ directly. But there is also a surprising forth outcome.

We have discovered and verified in laboratory experiments that energetic water ions can also participate in ER abstraction reactions on oxidized surfaces to directly form molecular $O_2$ anions. When oxygen is abstracted from such surfaces, it is readily replenished by (a) O fragments from the CID of $H_2O^+$ and (b) freshly exposed O atoms from physical sputtering of the mineral surface by $H_2O^+$. Thus, the ER reaction mechanism may generate $O_2^-$ continually on the cometary surface, which is emitted into the coma with high kinetic energy.

## Results

### $O_2^-$ and $HO_2^-$ production on cometary surface analogues.

We first demonstrate the production of molecular oxygen and hydroperoxyl radicals, detected as anions, from $H_2O^+$ ions bombarding Si, and Fe targets. These surfaces are covered with amorphous native oxide (hereafter $SiO_x$ and $FeO_y$), selected as analogues of two inorganic minerals commonly found on comets. Scattering on these surfaces produces multiple species from dissociation and physical sputtering, but also from direct reactions. Negative ion formation (Supplementary Figs 1 and 2) is of particular interest, since surface scattering has not been considered before as a production mechanism in cometary environments. Figure 1 shows energy distributions for $O_2^-$ and $HO_2^-$, scattered off from $SiO_x$ and $FeO_y$ as a function of the $H_2O^+$ incidence energy ($E_0$). For both surfaces, $O_2^-$ signal appears above $E_0 \sim 50$–60 eV. The peak position shifts to higher energy and the peak intensity goes through a maximum with increasing incidence energy. The scattering signal dies out above $\sim 200$ eV (Fig. 1a,c). The same is true for $HO_2^-$, though this peak appears earlier and dies out sooner than $O_2^-$ (Fig. 1b,d). These trends will

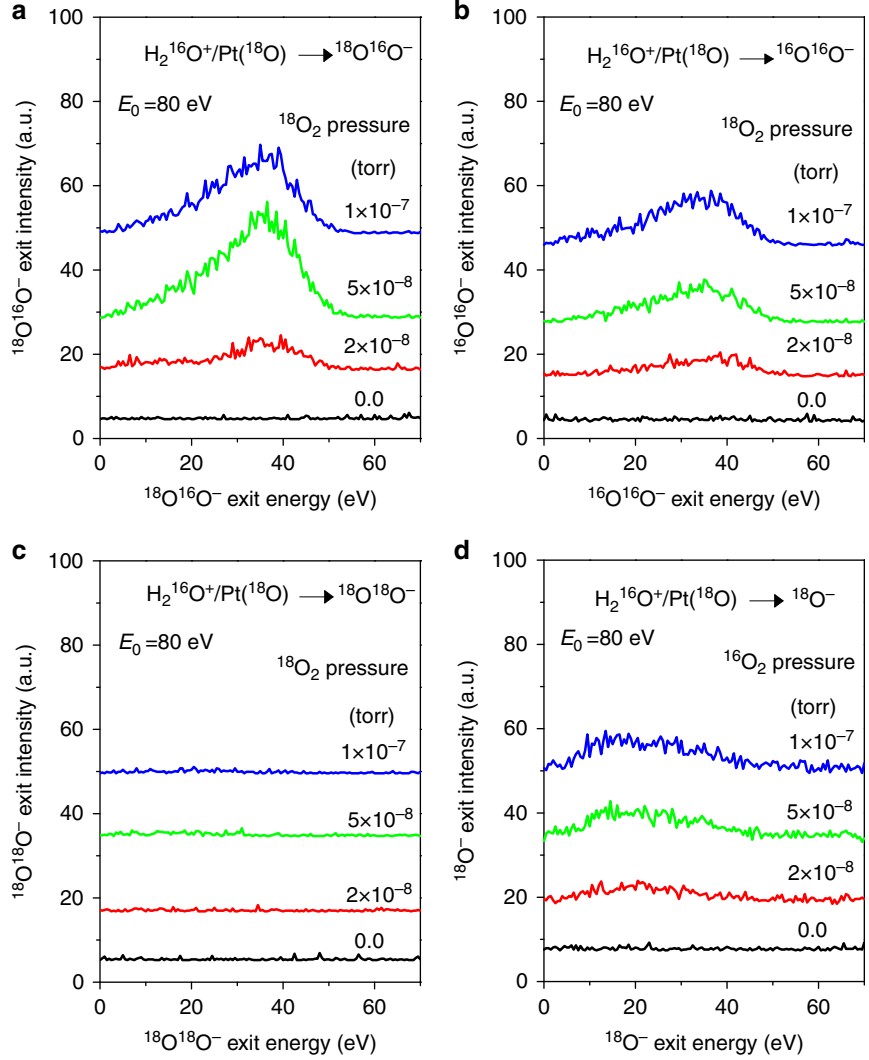

**Figure 2 | Isotopic dosing experiments of $H_2O^+$ scattering on Pt covered with $^{18}O$ atoms.** Energy distributions of ion exits of (**a**) $^{18}O^{16}O^-$, (**b**) $^{16}O^{16}O^-$, (**c**) $^{18}O^{18}O^-$ and (**d**) $^{18}O^-$ from $H_2O^+$ scattering on Pt at various $^{18}O_2$ exposure pressures, as annotated. The detection of $^{18}O^{16}O^-$ in **a** indicates fast $O_2^-$ formation between $^{16}O$-atoms from $H_2^{16}O^+$ and $^{18}O$-atoms adsorbed on the Pt surface. The absence of $^{18}O^{18}O^-$ in **c** proves that fast $O_2^-$ may not originate in the gas phase or from surface sputtering. The $^{16}O^{16}O^-$ formation in **b** is due to the oxygen deposition on the Pt surface from collision-induced dissociation of $H_2^{16}O^+$ (see text). The appearance of $^{18}O^-$ sputtering peak $\sim 25$ eV in **d** confirms that the surface is covered by $^{18}O$ atoms, produced by *in situ* $^{18}O_2$ exposure.

be understood below considering that $HO_2^-$ is a precursor to $O_2^-$. Are these anions sputtering products? Experiments with $Ne^+$ beams scattering on $SiO_x$ and $FeO_y$ surfaces indicate that $O_2^-$ is produced with intensity of $\sim 1$–2% of that seen for $H_2O^+$ at $E_0 = 110$ eV. Increasing the energy to $E_0 = 310$ eV, causes the $O_2^-$ signal to increase to 3% and 25% for $SiO_x$ and $FeO_y$, respectively, compared to $H_2O^+$ at the same energy (Supplementary Fig. 3). Under $^{18}O_2$ dosing, the sputtering contribution to $O_2^-$ signal increases to $\sim 15\%$ for $H_2O^+/FeO_y$, while undetectable for $H_2O^+/SiO_x$ at $E_0 = 110$ eV (Supplementary Fig. 4; Supplementary Table 1). Of course, sputtering is expected to increase for incidence energies above 300 eV, but the focus here is on much lower energies. With sputtering ruled out, the origin of $O_2^-$ is puzzling.

**Isotopic water scattering experiments on a model Pt surface.** Owing to the light atomic mass of Fe and Si, $H_2O^+$ scattering on cometary surface analogues produces dynamic $O_2^-$ peaks with energies over a limited range, with little separation from sputtering signal, thus hindering kinematic analysis. Collisional energy transfer is reduced on Pt, yielding larger spread in exit energies and much reduced overlap with sputtering peaks. Of course, the Pt surface needs to be oxidized first to render O-atom abstraction experiments possible. As mentioned above, CID of water ions is expected to supply O atoms to the Pt surface (proven below). However, neither $H_2O^+$ nor $D_2O^+$ ions, scattering on clean Pt, produce any discernible $O_2^-$ signal. This is likely due to molecular hydrogen, present in the ultrahigh vacuum background or formed *in situ* by an intramolecular water-splitting reaction[18], which scavenges O atoms from the Pt surface. The O-atom surface coverage must therefore be increased to overcome this problem, which is achieved readily by dosing the surface *in situ* with $O_2$ gas. Molecular $O_2$ dissociates spontaneously on Pt at room temperature[19], covering the surface with adsorbed O atoms—denoted hereafter as Pt(O).

Upon exposure of the Pt surface to $^{16}O_2$ gas, a strong dynamic $^{16}O_2^-$ peak appears for both $H_2^{16}O^+$ and $D_2^{16}O^+$ beams (Supplementary Fig. 5). What is the origin of the observed $^{16}O_2^-$? Isotopic $^{18}O_2$ dosing experiments on Pt, bombarded with normal $H_2^{16}O^+$ projectiles, can help distinguish possible contributions from sputtering or gas-phase collisions. Indeed, fast $^{18}O^{16}O^-$ and $^{16}O^{16}O^-$ peaks are detected, as shown in Fig. 2a,b. In contrast, there is no signal at 36 a.m.u. for all $^{18}O_2$ dosing pressures tried, confirming that no molecular $^{18}O^{18}O^-$ is produced (Fig. 2c). Since sputtered atomic $^{18}O^-$ is detected (Fig. 2d), the latter observation signifies that there is no $^{18}O_2$ on the Pt surface, whether adsorbed as a whole or formed by recombination of $^{18}O$ atoms. The formation of fast $^{18}O^{16}O^-$ is easily explained by an ER reaction of $H_2^{16}O^+$ abstracting adsorbed $^{18}O$. The formation of fast $^{16}O^{16}O^-$ may appear curious at first, as it requires the presence of $^{16}O$ on the surface. But it is readily accounted for by CID of $H_2^{16}O^+$ ions[18], which introduce $^{16}O$ or $^{16}OH$ to the Pt surface. As discussed above, $H_2^{16}O^+$ bombardment produces no $^{16}O^{16}O^-$ signal unless the Pt surface is dosed with molecular oxygen. Water dissociation alone cannot provide enough $^{16}O$-atom coverage owing to reactions with background hydrogen. Extra $^{18}O_2$ exposure reacts off such surface hydrogen, permitting beam-delivered $^{16}O$ to compete with $^{18}O$ atoms for surface sites.

Energy distributions for $^{18}O^{16}O^-$ (Fig. 3a), produced from direct abstraction of $^{18}O$ by $H_2^{16}O^+$ scattering on Pt($^{18}O$), are almost identical to those for $^{16}O^{16}O^-$, produced from $H_2^{16}O^+/$Pt($^{16}O$) (Fig. 3b). In both instances, the product $O_2^-$ peak position varies with $H_2^{16}O^+$ incidence energy, confirming further that the observed $O_2^-$ does not originate from sputtering. To avoid confusion, all subsequent references to oxygen—alone or in compounds—will be for $^{16}O$. Fast $O_2^-$ is also observed from $D_2O^+/$Pt(O) with similar incidence energy dependence (Fig. 3c). The peaks are narrow and well defined, providing accurate measurement of the exit energy. In addition to $O_2^-$, $H_2O^+/$Pt(O) produces scattered $H_2O^+$, $O_2^+$, $OH^{\pm}$, $O^{\pm}$, $H^+$ and $H_2^+$ (Supplementary Fig. 6). Surviving molecular $H_2O^+$ is only observed for $E_0 \leq 100$ eV. The $O_2^+$ signal is weak and noisy, indicating that $O_2$ mainly exits as $O_2^-$ (neutral $O_2$ is not studied). The kinematics of $OH^{\pm}$ and $O^{\pm}$ exits can be described by binary collision theory (BCT)[20], assuming that the collision produces an excited $H_2O$, which subsequently dissociates spontaneously (Supplementary Fig. 7).

**$O_2^-$ formation mechanism.** We propose that molecular $O_2$ forms by means of an ER reaction mechanism driven by energetic $H_2O$ collisions on oxidized surfaces (Fig. 4a). Following neutralization on approach, a transient state is formed at the distance-of-closest-approach (apsis) between the projectile ($H_2O$), the surface atom (S) and the adsorbate (O). Though short-lived, this state promotes a link between projectile and adsorbate. As the projectile rebounds, the transient state disintegrates, producing an energetic molecular product, $H_2O$–$O^{\star}$, in an excited state. We conjecture that this product is the elusive oxywater, a structural isomer of hydrogen peroxide and possible intermediate in oxidation reactions initiated by the latter[21]. The $H_2O$–$O^{\star}$ is unstable[22] and splits rapidly into a pair of ions, $H^+$ and $HO_2^-$, both of which are observed on $FeO_x$ and $SiO_y$ (Fig. 1). Weaker $HO_2^-$ signal is also observed for $H_2O^+/$Pt(O) (Supplementary Fig. 6i). The difference in scattered signal intensity from insulating versus metallic surfaces is noteworthy and important for determining $HO_2^-$ abundance in cometary comae (*vide infra*). The corresponding $H^+$ fragment possesses surprisingly high kinetic energy, larger than that estimated from mass-weighted energy partitioning of the $H_2O$–$O^{\star}$ parent. This is remarkable and must be contrasted with the second hydrogen atom, now on $HO_2^-$, which must also be ejected to produce $O_2^-$. This dissociation is promoted by remaining internal energy in the $HO_2^-$ fragment.

The proposed ER reaction mechanism is supported by an analysis of the collisional kinematics. The peak exit energies of $H_2O^+$, $O_2^-$ and $H^+$ product ions from $H_2O^+/$Pt(O) are summarized in Fig. 4b, as a function of the $H_2O^+$ incidence energy. Obviously, linear fittings capture the data well. Can the slopes be predicted? First, the kinematics of the $H_2O^+$ exit can be described very well by a slope of 0.8311, which is predicted by BCT when $H_2O^+$ scatters as a whole molecule. The intercepts of such fittings have been correlated to inelastic energy loss, termed 'inelasticity', associated with the production of a particular excited or ionic state[11]. The $H_2O^+$ inelasticity of $\sim 5$ eV seems small for the production of a highly excited state, so we ascribe this energy loss to a surviving water ion. The kinematics of the $O_2^-$ and $H^+$ products cannot be predicted by simple scattering arguments as they are fragments of an excited state, requiring knowledge of how the excitation energy is partitioned when the parent molecule breaks apart.

The critical element of the proposed mechanism is the oxywater transient state, which is not detected in our experiments. If it exists as a parent molecule, its energetics should be reflected in the daughter fragments, $H^+$ and $HO_2^-$. A simple summation of their measured kinetic energies may be used to estimate the energy of the $H_2O$–$O^{\star}$ parent, assuming late fragmentation. Given that $HO_2^-$ signal from scattering on Pt(O) is extremely weak, we must look at the $HO_2^-$ fragments, H and $O_2^-$, to estimate its energy. Since neutral H is not detected,

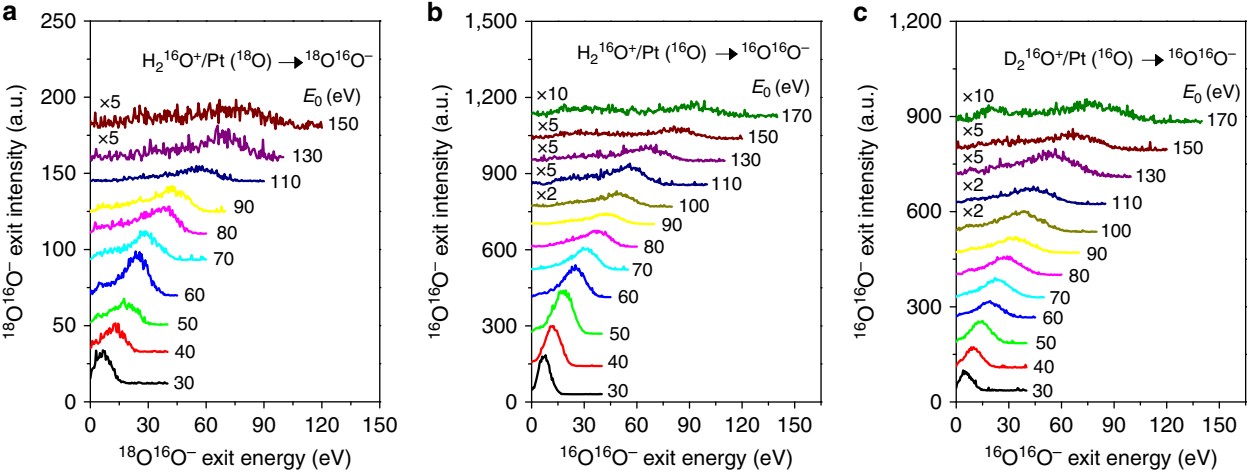

**Figure 3 | Direct formation of O$_2^-$ in collisions of normal water ions with an oxidized Pt surface.** Energy distributions of O$_2^-$ ions produced from (**a**) H$_2$O$^+$/Pt($^{18}$O), (**b**) H$_2$O$^+$/Pt($^{16}$O) and (**c**) D$_2$O$^+$/Pt($^{16}$O). The Pt surface was exposed to $^{18}$O$_2$ or $^{16}$O$_2$ *in situ* at a background pressure of $5 \times 10^{-8}$ torr. Results are shown for multiple incidence energies ($E_0$) of the corresponding water ions, as indicated. Very weak signal from sputtered O$_2^-$ appears as a second peak for $E_0 > 150$ eV.

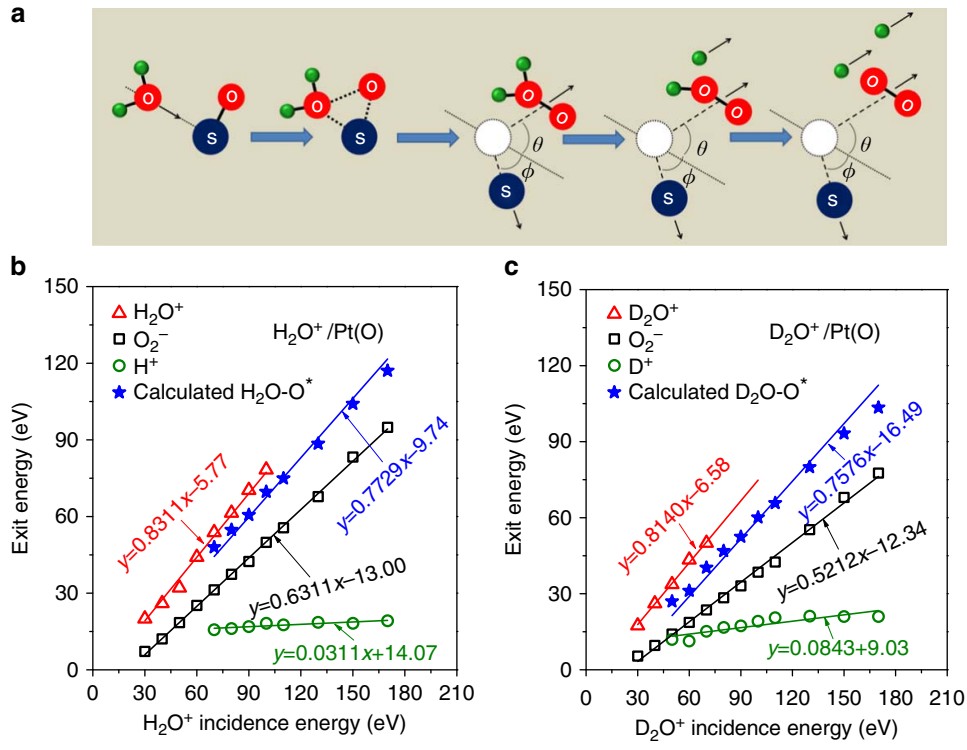

**Figure 4 | Proposed reaction mechanism and kinematics of direct O$_2$ formation from water.** (**a**) Schematic depiction of the proposed Eley–Rideal reaction mechanism between energetic water ions and adsorbed O-atoms, producing highly excited oxywater (H$_2$O-O* or D$_2$O-O*), which undergoes delayed fragmentation to form HO$_2$ (DO$_2$) as the precursor for O$_2$. (**b**) Ion exit energies of H$_2$O$^+$, O$_2^-$ and H$^+$ as a function of H$_2$O$^+$ incidence energy. The exit energy data of H$_2$O-O* were estimated from the measured exit energies of O$_2^-$ and H$^+$ (see text). (**c**) Ion exit energies of D$_2$O$^+$, O$_2^-$ and D$^+$ as a function of D$_2$O$^+$ incidence energy. The exit energy data of D$_2$O-O* were estimated (see text). All solid lines in **b,c** are linear fittings. The slopes for H$_2$O$^+$ and D$_2$O$^+$ are predicted from standard BCT. The slopes for H$_2$OO* and D$_2$OO* are calculated from a modified BCT model[11].

the HO$_2^-$ kinetic energy is estimated from the mass-weighted energy of O$_2^-$ fragment. Finally, the kinetic energy of H$_2$O-O* is obtained: $E(\text{H}_2\text{O-O*}) = 33/32 \cdot E(\text{O}_2^-) + E(\text{H}^+)$. This simple formula produces a number of points (Fig. 4b), which can be fitted relatively well by a straight line with a slope of 0.7729, corresponding to the kinematic factor predicted from BCT for H$_2$O–O* formation by an ER reaction[11].

The estimation method for the exit energy of the transient state is validated with data for D$_2$O$^+$/Pt(O). The peak exit energies of the D$_2$O$^+$, O$_2^-$ and D$^+$ product ions are summarized in Fig. 4c, as a function of the D$_2$O$^+$ incidence energy. Again, the kinematics of the D$_2$O$^+$ exit is described well by the predicted kinematic factor of 0.8140, when D$_2$O$^+$ scatters intact. The formula for calculating the D$_2$O–O* exit energy now becomes:

$E(D_2O-O^*) = 34/32 \cdot E(O_2^-) + E(D^+)$. The calculated points, shown in Fig. 3c, can be fitted by a straight line with a predicted[11] slope of 0.7576, though not as well as before.

The slopes of the data for $O_2^-$, $H^+$ (Fig. 4b) and $O_2^-$, and $D^+$ (Fig. 4c) cannot yet be predicted. Unconstraint, two-parameter linear fitting captures the data very well, suggesting a simple excited-energy partitioning mechanism between the oxywater molecule fragments. Comparing inelasticities, it appears that about the same energy is consumed to form $O_2^-$ from $H_2O^+$ versus $D_2O^+$. The $H^+$ and $D^+$ fittings (currently not understood) show positive inelasticity, implying an energy gain. Most of the internal energy of the oxywater transient state is likely converted into kinetic energy for the lighter fragments[23].

Similar ER reaction mechanisms yielding $O_2^-$ product also occur for $OH^+$, $OD^+$ and $O^+$ bombardment of Pt(O) (Supplementary Fig. 8). The $O_2^-$ exit energy data versus incidence energy for all incident ions can be described very well by BCT, confirming the validity of the kinematic analysis (Supplementary Fig. 9). The $OH^+$ and $O^+$ species are important because the 'accelerated water ions,' found in the 67P coma, include ions with molecular mass between 16 and 19 a.m.u. (ref. 12). Some of these $H_2O^+$ dissociation fragments are re-introduced into the coma, where the solar wind may pick them up and recycle them back to the comet surface, thus increasing $O_2$ production.

## Discussion

We have uncovered high-energy reaction channels for dynamic production of negative ions from collisions of energetic water ions with oxidized surfaces. The latter surfaces include: $SiO_x$, $FeO_y$, Pt(O), $NiO_z$, Pd(O), Au(O) and $TiO_w$ (see also Supplementary Figs 10–12). Such interactions are applicable to plasmas and astrophysical environments whenever $H_2O^+$ ions are encountered with kinetic energies between 50 and 300 eV. We propose that the scattering interactions occur in cometary comae during periods of activity, where they produce energetic negative ions, including: $O^-$, $OH^-$, $O_2^-$ and $HO_2^-$. The latter two ions, in particular, are produced by a novel ER reaction, and contribute to the $O_2$ abundance in the coma after photodetachment[24,25]. The lifetime of $O_2^-$ against photo-detachment is 2.6 s (at 1 a.u.)[25], which suggests that $O_2^-$ should be able to reach Rosetta. Negative ions should be present in the coma of 67P but they have yet to be reported, baring $H^-$ (ref. 26). Thus, our work actually predicts the existence of $O_2^-$ and $HO_2^-$ in the coma at distances sufficiently close to the nucleus to avoid photodetachment. Negative ions of cometary origin have been detected in the coma of comet 1P/Halley, though without sufficient mass resolution to distinguish individual ions[25]. Three broad peaks were observed, which were denoted as the 17-, 30- and 100-a.m.u. peaks. Chaizy et al.[24] argued that the first peak included $O^-$ and $OH^-$, while the second peak comprised $CN^-$. Furthermore, these authors considered several negative ion production mechanisms and found them inadequate to explain the signal intensity observed in the Halley coma. We propose that energetic water ion scattering off of the nucleus surface or dust grains in the coma of 1P/Halley could populate both the 17- and 30-a.m.u. peaks, and that the latter peak must have also included $O_2^-$ and $HO_2^-$.

The ER reaction mechanism is consistent with most reported[6] cometary $O_2$ observations and trends and must be an important contributor to its abundance. All necessary conditions for ER reactions on comet 67P are met: (i) water ions with the correct hyperthermal energies exist in the coma, and (ii) they impact the nucleus or dust grain surfaces, which (iii) contain oxidized materials. The mechanism explains well the strong correlation of $O_2$ to water abundance in the coma, and also the $O_2$ signal

increase closer to the nucleus—scattering makes the surface appear as a point source of $O_2$, thus justifying the observed $1/r^2$ dependence, where $r$ = cometocentric distance. The connection to solar wind may account for the relative invariance in the $O_2/H_2O$ ratio with heliocentric distance. That is, as the comet approaches the Sun, more subsurface sublimation leads to more water molecules in the coma; the solar wind also strengthens, increasing ionization and water ion flux to the surface, ultimately producing more $O_2$. The surface reaction is independent of the nucleus surface temperature or comet illumination conditions. The ER mechanism goes beyond these trends to explain the presence of significant amounts of $HO_2$ and the absence of $O_3$, which have baffled Bieler et al.[6] The findings are generic to comets irrespective of their origin in the early Solar System[27].

The primordial origin of cometary $O_2$ requires first a mechanism for $O_2$ formation. Water ice radiolysis by galactic cosmic rays during primordial times has been suggested[6] as that mechanism, despite evidence for very low $O_2$ abundance in protostellar envelopes[28]. Radiolysis is known to produce the chemically related species $O_3$, $H_2O_2$ and $HO_2$ (refs 29–31). The former two molecules are stable and should have also been incorporated into the comet at the same time as $O_2$. However, no $O_3$ has been detected in the 67P coma, a concern identified by Bieler et al.[6] that also applies to other efforts at explaining the primordial origin[32–34]. On the other hand, $H_2O_2$ and $HO_2$ have been detected and their gaseous abundance ratios were reported for the 67P coma: $H_2O_2/O_2 = 0.6 \times 10^{-3}$ and $HO_2/O_2 = 1.9 \times 10^{-3}$. These were compared with the abundance ratios measured in the $\rho$ Oph A dense core[29], where $O_2$ has also been detected and is likely to originate from radiolysis: $H_2O_2/O_2 \approx HO_2/O_2 \approx 0.6 \times 10^{-3}$. The $H_2O_2$ abundance relative to $O_2$ is clearly a perfect match but the value for the $HO_2/O_2$ ratio is $3\times$ larger in the coma of 67P. This difference suggests that $HO_2$ is formed at higher rates than it can be destroyed, thus accumulating in the coma. It is likely that $HO_2$ forms by a mechanism different from or perhaps in addition to that operating under interstellar conditions. Apart from the mechanism discussed in this communication, there is actually another reaction mechanism enabled by the presence of $O_2$ in the coma, which involves a different ER reaction[35]. Like $H_2O^+$, photo-ionized $O_2^+$ can be picked up by solar wind and accelerated back to the comet, where it can abstract atomic H from cometary materials to form $HO_2$.

There have been other attempts to justify the primordial $O_2$ formation and its survival for 4.6 billion years. Mousis et al.[32] considered the radiolysis of icy grains in the low-density proto-solar nebula, which may produce large amounts of $O_2$ though 'its incorporation as crystalline ice is highly implausible'. These authors discussed two extreme $O_2$ production scenarios for dense and early proto-solar nebula, which require very large galactic cosmic ray fluxes and $O_2$ trapping in clathrates. Taquet et al.[33] used sophisticated astrochemical models to compare various primordial $O_2$ formation mechanisms, proposing oxygen atom recombination at the surface of interstellar ices as a possibility, albeit under 'warmer and denser conditions than usually expected in dark clouds'. Finally, Dulieu at al.[34] proposed that $O_2$ forms in situ during the evaporation of water ice via a dismutation reaction of co-evaporating $H_2O_2$. This mechanism requires the incorporation of primordial $H_2O_2$ in large amounts into the nucleus and its complete conversion into $O_2$ to be consistent with the low levels of $H_2O_2$ in the coma. All these mechanisms appear to be in conflict with the relative abundances of the related species $O_3$, $H_2O_2$ and $HO_2$.

In contrast, the ER reaction mechanism comprises cometary ions and minerals actually found on 67P. It produces $O_2$ in situ through the putative oxywater state, which dissociates

spontaneously into $HO_2^-$ as an intermediate on its way to $O_2^-$ or $HO_2$ following photodetachment. That is, $HO_2$ is co-produced by this new reaction and should add to any amounts formed by other mechanisms. It is reasonable then to expect that the $HO_2$/$O_2$ ratio should be larger than interstellar values[30]. The remaining factor, $H_2O_2$, can form collisionally in the extended coma by hydrogen atom transfer from scattered $HO_2^-$ or $HO_2$ to ambient $H_2O$. Finally, no $O_3$ is produced during the ER reaction, consistent with observations.

The crux of any proposed $O_2$ production mechanism is whether it can explain the observed $O_2$ abundance in the 67P coma. The $O_2^-$ production rate by the ER reaction is proportional to the accelerated water ion flux, which has been measured to be $3 \times 10^9$–$3 \times 10^{11}$ $m^{-2}$ $s^{-1}$ at 2 a.u. with the caveat that it may be underestimated by at least two orders of magnitude[13,36]. The proportionality constant ($O_2$ yield) cannot be estimated at this time. Regardless, the reported flux is too low to make 'accelerated' water ions entirely responsible for the reported $O_2$ abundance. However, it is noteworthy that there are other water-derived ions present in the coma, which can participate in similar ER reactions on the nucleus surface. For example, the extended coma also contains abundant $H_3O^+$, $OH^+$ and $O^+$ (ref. 36), all of which can be picked up by the solar wind and accelerated to energies sufficient to drive ER reactions (Supplementary Figs 8 and 9). Unfortunately, the flux and energy distributions of these additional ions have not been reported—$H_3O^+$, in particular, may be a serious contributor as its density in the inner coma varies considerably and can reach 100 times the density of $H_2O^+$ (ref. 37). Furthermore, the 'cold' water ions[5] have two orders of magnitude larger flux and possess kinetic energies up to 50 eV but move away from the comet, so they can produce $O_2$ only in collisions with dust grains. Without some knowledge of these additional species flux and energy distributions and of the dust grain density in the coma, we cannot quantify the magnitude of the ER reaction contributions to the cometary $O_2$.

Nevertheless, we note that a unique feature of the ER reaction mechanism is its ability to produce energetic $O_2^-$ anions, moving away from the nucleus towards the orbiting Rosetta spacecraft with kinetic energy between 10 and 50 eV. The Rosetta double focus mass spectrometer (ROSINA/DFMS) entrance slit plate is biased in gas mode to reject ambient ions[38]. Positive bias will attract and accelerate $O_2^-$ into the ionizer box. Depending on cometocentric distance, some or all of the $O_2^-$ anions in transit to Rosetta will undergo photodetachment[24], producing neutral $O_2$ molecules, which retain their kinetic energy and will enter the ionizer regardless of bias. Energetic collisions of hyperthermal $O_2^-$ ions or neutralized $O_2$ molecules with the gold-coated internal surfaces of the ROSINA/DFMS ionizer will produce $O_2^+$ ions by surface re-ionization[39] (Supplementary Fig. 11). It is worth pondering how $O_2^+$ formed inside the ROSINA/DFMS by a mechanism other than electron impact ionization will contribute to the detected $O_2^+$ signal[40].

In conclusion, energetic water ions in cometary comae, produced and accelerated by solar wind, can drive scattering interactions on cometary surfaces that alter the relative speciation in the coma. CID of water ions on the comet can generate negative ions ($O^-$ and $OH^-$). Abstraction of chemisorbed oxygen from oxidized surfaces by water ions can also produce dynamically $O_2^-$ and $HO_2^-$, by means of a previously unknown ER reaction. Kinematic analysis of the ER reaction products provides indirect evidence for the elusive oxywater state as a reaction intermediate, which may form during the hard collision, then dissociate promptly on the rebound from the surface. When this reaction occurs in comets, it can populate the coma with energetic $O_2^-$ anions, which are converted readily to molecular $O_2$ by photo-detachment. This abiotic way to produce molecular $O_2$ informs our understanding of cometary chemistry and could be important in other astrophysical environments.

## Methods

**Materials.** Polycrystalline Pt and Fe foils (4 N purity, ESPI) were used as received or sputter-cleaned *in situ* as needed. Doped Si wafers (n-type) covered with native silicon oxide (thickness 2–3 nm) were degreased but otherwise left untreated. Research grade $^{16}O_2$ and $^{18}O_2$ gases (5 N) were used for the dosing experiments.

**Scattering apparatus operation.** Laboratory experiments were carried out in an ultra-high vacuum ion scattering system connected to an ion beam line, as described in detail elsewhere[41,42]. Positive ions were extracted from an inductively coupled plasma of $H_2O$ or $D_2O$ in Argon carrier gas, operated at 5 mtorr and 500 W of radio-frequency power supplied at 13.56 MHz. The ions were launched into the ion beam line at − 15 KV and were magnetically mass-filtered to produce isotopically pure beams of $O^+$, $OH^+$, $OD^+$, $H_2O^+$ and $D_2O^+$, with fluxes between 2 and 5 µA. Ion energy was tuned by adjusting the plasma potential with respect to ground using external bias. The energy width of all incident beams was constant at ~5 eV (full width at half maximum), determined by the electron temperature in the plasma. The ion beams were then decelerated and delivered to grounded target surfaces, held at room temperature. The beam waist on the sample was ~3 mm. Gas dosing of the target surfaces was accomplished using a tube situated ~2 cm from the surface. O-atom coverage was adjusted (but not measured) by changing the background $O_2$ pressure using a leak valve. The ions impinged on the surface at 45° angle of incidence and the scattered products were detected at 45° angle of exit in the scattering plane. Though some incident ions survive the collision as charged species, most are typically neutralized efficiently on the incoming trajectory via charge exchange with the surface[43]. The violent collision with the metal surface causes hyperthermal surface ionization[39], a process that allows some of the scattered products to exit the surface as positive or negative ions without resorting to electron-impact or photo-ionization schemes. In fact, studies of scattering at high energies would not be possible without surface ionization, as most conventional electron-impact ionizers are very inefficient at high product exit velocities. It is for this reason that no neutral products could be resolved in continuous wave experiments; their presence has been verified in calibration experiments by pulsing the beam and using lock-in detection schemes. All charged scattered products were chemically identified with a high-resolution mass spectrometer (Extrel QPS) and their translational energies were measured using a calibrated 90°-sector energy analyser. All signals reported were normalized to the corresponding beam current.

**Cometary surface analogues.** Thin oxides are used as analogues of cometary materials in order to avoid localized surface charging, which occurs during beam exposure at high ion flux. Surface charging interferes with the measurement of ion exit energies. When using native silicon or iron oxide (~2–3 nm), electron tunnelling to the underlying conductive substrate allows removal of the surface charge. The presence of solar wind in the cometary environment mitigates this problem, by providing electrons to the exposed cometary materials (silicates, quartz and so on) to neutralize this charge. The choice of the oxides is obvious: (1) oxygen atoms on the silicon oxide surface are bonded to Si in a similar fashion to surface oxygen atoms in silicates, and (2) elemental Fe has been found on 67P, and given the oxidizing conditions in the coma, oxidized Fe surfaces are to be expected. Furthermore, the charging effect has been verified by scattering on titanium oxide, which is a semiconductor and has mild conductivity, but exhibits the same scattering behaviour as silicon oxide (Supplementary Fig. 12).

**Data availability.** All relevant data supporting the findings of this study are available from the corresponding author upon request.

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

## Acknowledgements

This report was based on work funded by the National Science Foundation/Department of Energy Partnership for Basic Plasma Science and Engineering (Award No. 1202567).

## Author contributions

Y.Y. and K.P.G. designed the experiments and co-wrote the paper. Y.Y. conducted experimental measurements. K.P.G. supervised the project.

## Additional information

**Competing interests:** The authors declare no competing financial interests.

