## [Peer Review File · Nature Communications]

Reviewers' comments:

Reviewer #1 (Remarks to the Author):

The paper "Dynamic molecular oxygen production in cometary comae" by Dr. Giapis and colleagues describes a new laboratory experiment, which allows producing O₂ from a surface interacting with water ions. The surface is assumed to be a cometary analogue, and hence results from this experiment may be of peculiar importance to investigate the presence of O₂ molecules in cometary comae, as for instance observed in comet 67P/C-G. The results apply to other comets as well as to other small bodies in the Solar System.

The results are new and of interest for planetary science in general.
The laboratory experiment is described with enough details in 'Methods'.

An important aspect to face in more details is the dimension of oxidized inclusions on the comet's surface (or grains) that are required to generate a sufficient flux of O₂.
Do you have an estimation of the oxides deposit dimensions on the substrate? Are these regions compatible with the inclusions inferred in the comet's surface?

In addition, it would be interesting to add a brief discussion of the results in the framework of comet 67P, including for instance Energy/speed of produced molecules values and the water ions and products fluxes in comparison with the Rosetta outcomes.

This would integrate the paper with an example of current interest for the scientific community.

With minor revision, the paper is recommended for publication on Nature Communication.

In the following some detailed remarks to the text:

- page 1, line 19: 'is HO₂, also present in the coma of 67P at high levels'...
Please specify the meaning of this sentence and/or please provide a reference to support this statement.

- page 2, line 42-44
Please provide information about the dimension of the simulated surface and oxides.

- page 2, line 45: 'Such scattering produces multiple ...'
Not clear. I would suggest changing with: 'We notice that this scattering produces in addition multiple species from both reactions and sputtering (figs. S1, S2)' (or a similar sentence)

- page 5, line 97: Fig 1
Results refer also to figs. S1 and S2. Please add.

- page 6, lines 121 and 128:
Is there a reference for the reported formulas? Or, please provide a short description.

- page 7, lines 161 and 163:
The statement is not clear: the water ion fluxes are available for a heliocentric distance of 3.2 to 2.8 AU in the cited paper (values of 10⁸-10¹⁰ m⁻² s⁻¹ as reported by Nilsson et al., Science, 2015). O₂⁻ cannot be disentangled from water ions, and hence its flux is not directly measured (see Fuselier et al., A&A, 2015).

Do you have an estimation of the O₂⁻ density? Is it consistent with the expected values from Rosetta experiments?

References:

Reference # 20 (in the reference list and on page 7): maybe you were referring to the following paper:

Nilsson et al., Evolution of the ion environment of comet 67P/Churyumov-Gerasimenko. Observations between 3.6 and 2.0 AU, Astronomy and Astrophysics, 583, A20, (2015).

Figure 3b: is there an error in the reaction? Should it be $\text{H}_2\text{16O}^+/\text{Pt}(16\text{O}) \rightarrow 16\text{O}16\text{O}^-$ instead of $16\text{O}16\text{O}_2^-$?

Reviewer #2 (Remarks to the Author):

The in-situ detection of a high ratio of $\text{O}_2:\text{H}_2\text{O}$ (4%) during the ROSETTA mission have been a huge surprise for the astrophysical community (Bieler et al 2015). The authors wrote "The O_2 origin was ascribed to primordial gaseous O_2 incorporated into the nucleus during the comet's formation []. This thesis was put forward after carefully considering an exhaustive list of known O_2 production mechanisms in cometary environments, including photolysis and radiolysis of surface water ice, solar wind-surface interactions, and gas-phase collisions" .

Among other possible mechanisms to form O_2 on cometary environments, the authors propose a new possibility of ER reaction of accelerated H_2O^+ ions with the oxidized surface of comets or grains. However, there is no information about the efficiency of the process. The other mechanisms have been ruled out for quantitative reasons, not because they were not able to produce O_2 , but because they did not produce enough O_2 . Authors wrote themselves "The ER reaction mechanism explains most observations about O_2 in the 67P coma, albeit not its reported high abundance "

Remark 1 : if this mechanism was known at this time, would have it been ruled out by Bieler et al ? More precisely, please give an order of magnitude of the O_2 production via this mechanism. It is the weakest point of the article. Naïvely, I consider that the degree of ionization is such that obtaining few % of the production of the neutral gas via energetic ion colliding with surfaces is extremely optimistic.

Remark 2: Since the discovery of high amount of O_2 , other scenarios have been explored, especially chemical routes. They should be discussed in the paper : Dulieu et al 2017, Taquet et al 2016.

Oxydized Fe, Si, and Pt surfaces are used to mimic olivine or pyroxene silicates or non volatile (organic) refractory materials. Actually, the ER mechanism seems to be transportable from SiO_x , FeO_x to $\text{Pt}(\text{O})$, but the generalization to cometary surfaces is unclear to me.

Question : Is it reasonable to think that any oxydised surface materials (quartz) would give similar results ?

Remark 3: The choice of these surfaces and their astrophysical relevance has to be discussed, and I wonder why silicates was not used for example.

Experimental methods and section: As far as I understood, the authors demonstrate that H_2O^+ bombardment of oxydised Fe surfaces Si surfaces and $\text{Pt}(\text{O})$ produce HO_2^- which evolve into O_2^- . They discuss about the possible $\text{H}_2\text{O}-\text{O}^*$ intermediate.

It is absolutely unclear that they have ever detected any O_2 molecules or, that, taken into account that ions are usually more easily detected, that there is an indirect method to conclude that O_2 is directly formed. On the contrary, it is clear that O_2^- have high kinetic energy that can be measured.

My understanding is that mostly O_2^- is produced, and not O_2 .

Remark 5: Please clarify this point. It is not correct to let the reader think that the paper is about O_2 if O_2^- is the real product and object of study.

Remark 4: Because O_2^- is a key tracer of the ER mechanism, a careful attention should be paid to the presence of this anion in cometary atmospheres. To my knowledge, no O_2^- detection have been reported, whereas others anions (H^- , OH^- CN^-) have been reported (ref 21). It is certainly a point to discuss, especially if no O_2^- have been detected.

Line 161-171 (concluding paragraph): I don't really understand what do the authors mean. It is very loose. "some or all O_2^- may be neutralized by photodetachment" contains no information but is central, if orders of magnitude are provided. O_2^- has a relatively low electron affinity, therefore, its photodetachment rate should be high under solar light (IR and Visible). It is thus reasonable to think that 50eV neutral O_2 will rapidly escape from the atmosphere or strongly interact with it. All these aspects should be discussed more carefully in order to make the link between the experiments and the observations.

General conclusion:

The authors performed a valuable new piece of science, and they show how H_2O^+ (and similar impactors) in the energy range of 100eV can produce HO_2^- which evolves into O_2^- of few tens of kinetic energy for the larger part of the incident energies, after abstraction of an oxygen atom of oxidized surfaces of Fe Si and Pt(O). They clearly show that it is not simple sputtering but really the ER mechanism. However, even if "all necessary conditions for such reactions are met are present in the cometary atmosphere" (and we can discuss the point of the surfaces), I found that in there is no key to understand how much this process can explain the in-situ measurement of Bieler et al. Moreover, I found the gap between the title and the experiments which support the real scientific discussion very deep. The contrast between the high level of the experimental discussion and the approximative astrophysical arguments is obvious.

Yes, it is reasonable to think that the astrophysical scenario proposed here may contribute to the production of O_2 . Is it the real conclusion of the authors ? Actually I don't know, I don't understand what is their final conclusion.

At this stage I suggest the authors to choose whether their claim is an ambitious astrophysical scenario and therefore to convince with adequate order of magnitudes that this scenario is solid, or to focus on experiments, as they actually mostly did, but thus to better show where is the real novelty. I also suggest to be clear about the role of anions.

Question related to remark 1: Please provide explicitly the $\text{H}_2\text{O}^+/\text{H}_2\text{O}$ ratio in cometary atmosphere (or a range because it is space and time dependent). Give an order of magnitude of the total efficiency of the reaction (How many O_2 per H_2O^+ sent to the surface).

Suggestion : What about the proposed mechanism and the elusive detection of O_2 in the ISM ?

Reviewer #3 (Remarks to the Author):

It has heretofore not been possible to explain the large amount of O_2 found in the coma of comet 67P using standard theories and mechanisms. The authors provide experimental evidence for the production of O_2 via a previously unexplored Eley-Rideal mechanism involving cationic water collisions with oxide surfaces. Gas-surface scattering experiments are performed in UHV involving high-energy ion beams with the detection of scattered ions via mass spec.

H_2O^+ is scattered from amorphous Si and Fe oxide surfaces over a wide range of energies, and shown to produce O_2^- and HO_2^- . The authors make a reasonable argument against the formation of O_2^- via sputtering. Studies of H_2O^+ scattering from O-covered Pt surfaces, using different isotopic combinations for the O atoms, are consistent with this argument. The authors propose that the incident H_2O^+ abstracts a chemisorbed O atom from the oxide surface (an Eley-Rideal reaction) to form oxywater, H_2O_2 . The oxywater in turn rapidly dissociates into H^+ and HO_2^- , and the HO_2^- fragment eventually dissociates into H and O_2^- . Unfortunately, there is no direct evidence for the formation of the H_2O_2 . However, its existence has been argued using high-level ab initio theory, and it is reasonable that any H_2O_2 formed via this pathway would be very highly excited, and would likely break into more stable products. Both H^+ and HO_2^- are detected in the experiments, and in the comet. The most compelling evidence comes from the kinematics of the scattered O_2^- and H^+ . The energies of these scattered products are consistent with the proposed Eley-Rideal mechanism, using standard binary collision theory to compute the scattering energy of the H_2O_2 parent.

I cannot comment on the astrophysical or experimental merits of this paper, but the proposed Eley-Rideal mechanism appears to be plausible, new, interesting, and consistent with the available data. Obviously much work needs to be done, but I think this paper is suitable for publication.

Response to Reviewers' comments

Reviewer #1 (Remarks to the Author):

The paper “Dynamic molecular oxygen production in cometary comae” by Dr. Giapis and colleagues describes a new laboratory experiment, which allows producing O₂ from a surface interacting with water ions. The surface is assumed to be a cometary analogue, and hence results from this experiment may be of peculiar importance to investigate the presence of O₂ molecules in cometary comae, as for instance observed in comet 67P/C-G. The results apply to other comets as well as to other small bodies in the Solar System.

The results are new and of interest for planetary science in general.
The laboratory experiment is described with enough details in ‘Methods’.

An important aspect to face in more details is the dimension of oxidized inclusions on the comet’s surface (or grains) that are required to generate a sufficient flux of O₂.

Do you have an estimation of the oxides deposit dimensions on the substrate? Are these regions compatible with the inclusions inferred in the comet’s surface?

ANSWER: The ER reaction mechanism that produces O₂ is generic to the surface so long as oxygen is present. We have found this to be true for: SiO_x, FeO_x, NiO_x, TiO_x, Pt(O), Pd(O), and Au(O), see Suppl. Info. Even if the surface is not oxidized, water ions can provide oxygen after undergoing collision-induced dissociation (CID). For the same reason, surface oxygen removed by the ER reaction can be replenished. Since there are ions with higher energies (up to 800eV) in the coma, physical sputtering can uncover fresh oxygen in oxidized minerals. Thus, even if oxidized inclusions were small to begin with, the whole sample surface can become oxidized by the incident energetic water ions. On page 2, we added the following clarification sentence:

The outer crust of the 67P nucleus facing the sun is dehydrated¹⁰ thus exposing mineral surfaces to the ions. When oxygen is abstracted from such surfaces, physical sputtering by the water ions can expose more oxygen below. In addition, CID of water ions provides another in situ source of oxygen to replenish abstracted oxygen.”

In addition, it would be interesting to add a brief discussion of the results in the framework of comet 67P, including for instance Energy/speed of produced molecules values and the water ions and products fluxes in comparison with the Rosetta outcomes.

This would integrate the paper with an example of current interest for the scientific community.

ANSWER: We have revised the discussion and added new text to emphasize our results in the context of cometary observations, including comets 67P and 1P/Halley. Please see highlighted portions on pages 7-9 of the revision. The Discussion Section has now the following headings:

Discussion and implications for cometary measurements

Negative ion production.

O₂ trends in the 67P coma

Toward an explanation of the cometary O₂ abundance

With minor revision, the paper is recommended for publication on Nature Communication.

In the following some detailed remarks to the text:

- page 1, line 19: 'is HO₂, also present in the coma of 67P at high levels' ...

Please specify the meaning of this sentence and/or please provide a reference to support this statement.

ANSWER: Nature Communications does not permit citing of references in the abstract. We have removed the offending sentence from the abstract and added the following text in the Discussion on page 8:

The primordial origin of cometary O₂ requires first a mechanism for O₂ formation. Water ice radiolysis by galactic cosmic rays during primordial times has been suggested²¹ as that mechanism, despite evidence for very low O₂ abundance in protostellar envelopes²³. Radiolysis is known to produce the chemically related species O₃, H₂O₂, and HO₂ (refs 24-26). The former two molecules are stable and should have also been incorporated into the comet at the same time as O₂. However, no O₃ has been detected in the 67P coma, a concern identified by Bieler et al. that also applies to other efforts at explaining the primordial origin²⁷⁻²⁹. On the other hand, H₂O₂, and HO₂ have been detected and their abundance ratios were reported for the 67P coma: H₂O₂/O₂ = 0.6x10⁻³ and HO₂/O₂ = 1.9x10⁻³. These were compared with the abundance ratios measured in the ρ Oph A dense core²⁴, where O₂ has also been detected and is likely to originate from radiolysis: H₂O₂/O₂ ≈ HO₂/O₂ ≈ 0.6x10⁻³. The H₂O₂ abundance relative to O₂ is clearly spot on but the value for the HO₂/O₂ ratio is 3x larger in the coma of 67P. We argue that this ratio should be either 0.6x10⁻³ (i.e., HO₂ has survived intact in the ice), or lower; it should not be larger as HO₂ cannot be formed to any significant degree through gas phase collisions of O₂ and H in the tenuous coma. This discrepancy is significant and, together with the absence of O₃, constrains the primordial radiolysis conjecture.

- page 2, line 42-44

Please provide information about the dimension of the simulated surface and oxides.

ANSWER: We have revised the Methods section to include the sentence:

The ion beams were then decelerated and delivered to grounded target surfaces, including surface-oxidized Fe, Si, (SiO_x thickness ~3nm) and oxygen-covered polycrystalline Pt surfaces (4N purity, ESPI), held at room temperature. The beam waist on the sample was approximately 3 mm.

- page 2, line 45: 'Such scattering produces multiple ...'

Not clear. I would suggest changing with: 'We notice that this scattering produces in addition multiple species from both reactions and sputtering (figs. S1, S2)' (or a similar sentence)

ANSWER: We have revised this sentence on page 3 to read:

Scattering on these surfaces produces multiple species from dissociation, physical sputtering, but also direct reactions. Negative ion formation (Supplementary Fig.1, 2) is of particular interest, as surface scattering has not been considered before as a production mechanism in cometary environments.

- page 5, line 97: Fig 1

Results refer also to figs. S1 and S2. Please add.

ANSWER: These figures (S1, S2) can be found in the Supplementary Information file.

- page 6, lines 121 and 128:

Is there a reference for the reported formulas? Or, please provide a short description.

ANSWER: This is a simple calculation that does not require a reference. When a large molecule dissociates in the gas-phase, its fragments move with the same velocity. Thus, the kinetic energy of the initial molecule is split between the fragments according to their mass. This is simple energy conservation.

- page 7, lines 161 and 163:

The statement is not clear: the water ion fluxes are available for a heliocentric distance of 3.2 to 2.8 AU in the cited paper (values of 10^8 - 10^{10} $\text{m}^{-2} \text{s}^{-1}$ as reported by Nilsson et al., Science, 2015). O_2^- cannot be disentangled from water ions, and hence its flux is not directly measured (see Fuselier et al., A&A, 2015).

Do you have an estimation of the O_2^- density? Is it consistent with the expected values from Rosetta experiments?

ANSWER: The O_2^- flux produced by the accelerated water ions is probably several orders lower than that reported by Bieler et al. There are, however, contributions from other species found in the coma (e.g., H_3O^+ , OH^+ and O^+), which react in a similar fashion to produce O_2 . Without some knowledge of their flux and energy distributions, we cannot provide an estimate of the magnitude of the ER reaction contribution to cometary O_2 . We have added the following text on page 9:

The O_2^- production rate by the ER reaction is proportional to the accelerated water ion flux, which has been measured to be 3×10^9 - 3×10^{11} m^{-2}/s at 2 AU with the caveat that it may be underestimated by at least two orders of magnitude^{5,31}. This flux is too low to make “accelerated” water ions entirely responsible for the reported O_2 abundance. The extended coma also contains abundant H_3O^+ , OH^+ and O^+ (ref. 31), all of which can be picked up by the solar wind and accelerated to energies sufficient to drive similar ER reactions on the nucleus surface (Suppl. Figs. S8, S9). However, the flux and energy distributions of these additional ions have not been reported. Furthermore, the “cold” water ions⁵ have two orders of magnitude larger flux and kinetic energies up to 50 eV but move away from the comet, so they can produce O_2 only in collisions with dust grains. Without some knowledge of these additional species flux and energy distributions, and dust grain density in the coma, we cannot quantify the magnitude of the ER reaction contributions to the cometary O_2 .

References:

Reference # 20 (in the reference list and on page 7): maybe you were referring to the following paper:

Nilsson et al., Evolution of the ion environment of comet 67P/Churyumov-Gerasimenko. Observations between 3.6 and 2.0 AU, Astronomy and Astrophysics, 583, A20, (2015).

ANSWER: We have added this reference in the revised manuscript as Ref.5.

Figure 3b: is there an error in the reaction? Should it be $\text{H}_2\text{16O}^+/\text{Pt}(16\text{O}) \rightarrow 16\text{O16O}^-$ instead of 16O16O_2^- ?

ANSWER: The error in Figure 3b has been corrected.

Reviewer #2 (Remarks to the Author):

The in-situ detection of a high ratio of O₂:H₂O (4%) during the ROSETTA mission have been a huge surprise for the astrophysical community (Bieler et al 2015). The authors wrote “The O₂ origin was ascribed to primordial gaseous O₂ incorporated into the nucleus during the comet’s formation [1]. This thesis was put forward after carefully considering an exhaustive list of known O₂ production mechanisms in cometary environments, including photolysis and radiolysis of surface water ice, solar wind-surface interactions, and gas-phase collisions” .

Among other possible mechanisms to form O₂ on cometary environments, the authors propose a new possibility of ER reaction of accelerated H₂O⁺ ions with the oxidized surface of comets or grains. However, there is no information about the efficiency of the process. The other mechanisms have been ruled out for quantitative reasons, not because they were not able to produce O₂, but because they did not produce enough O₂. Authors wrote themselves “The ER reaction mechanism explains most observations about O₂ in the 67P coma, albeit not its reported high abundance “

Remark 1 : if this mechanism was known at this time, would have it been ruled out by Bieler et al ? More precisely, please give an order of magnitude of the O₂ production via this mechanism. It is the weakest point of the article. Naïvely, I consider that the degree of ionization is such that obtaining few % of the production of the neutral gas via energetic ion colliding with surfaces is extremely optimistic.

ANSWER: The mechanism of O₂ formation via ER reactions is new and surprising. Until the Rosetta mission, it was not known that “accelerated” water ions could be found in cometary comae and that they could collide with the nucleus. Since such ions exist and impact the comet, they will produce O₂. We are unable to quantify the production rate at this point. We have revised the manuscript on page 9 to explain why:

The O₂⁻ production rate by the ER reaction is proportional to the accelerated water ion flux, which has been measured to be $3 \times 10^9 - 3 \times 10^{11}$ /m²/s at 2 AU with the caveat that it may be underestimated by at least two orders of magnitude^{5,31}. This flux is too low to make “accelerated” water ions entirely responsible for the reported O₂ abundance. The extended coma also contains abundant H₃O⁺, OH⁺ and O⁺ (ref. 31), all of which can be picked up by the solar wind and accelerated to energies sufficient to drive similar ER reactions on the nucleus surface (Suppl. Figs. S8, S9). However, the flux and energy distributions of these additional ions have not been reported. Furthermore, the “cold” water ions⁵ have two orders of magnitude larger flux and kinetic energies up to 50 eV but move away from the comet, so they can produce O₂ only in collisions with dust grains. Without some knowledge of these additional species flux and energy distributions, and dust grain density in the coma, we cannot quantify the magnitude of the ER reaction contributions to the cometary O₂.

However, our mechanism is consistent with all other trends and dependencies reported by Bieler et al. We discuss in the text (Page 7-8) why our mechanism explains the cometocentric and heliocentric distance dependencies. We can explain why comet surface illumination does not affect the production of O₂. Furthermore, we can explain two additional observations that have baffled Bieler et al.:

- 1) Bieler et al. cannot explain why there is no O₃ detected. The primordial O₂ hypothesis requires that it be present in the coma of 67P in detectable amounts.

- 2) Bieler et al. compares the H₂O₂:O₂ and HO₂:O₂ ratios measured on 67P to those measured in the ρ Oph A dense cloud, where O₂ has been found and is likely due to radiolysis. While the H₂O₂:O₂ ratio is spot on, the HO₂:O₂ ratio is **3x larger** in 67P. Bieler et al. do not discuss this discrepancy.

We have added the following paragraph to the Discussion Section of the revision (Page 8):

The primordial origin of cometary O₂ requires first a mechanism for O₂ formation. Water ice radiolysis by galactic cosmic rays during primordial times has been suggested²¹ as that mechanism, despite evidence for very low O₂ abundance in protostellar envelopes²³. Radiolysis is known to produce the chemically related species O₃, H₂O₂, and HO₂ (refs 24-26). The former two molecules are stable and should have also been incorporated into the comet at the same time as O₂. However, no O₃ has been detected in the 67P coma, a concern identified by Bieler et al. that also applies to other efforts at explaining the primordial origin²⁷⁻²⁹. On the other hand, H₂O₂ and HO₂ have been detected and their abundance ratios were reported for the 67P coma: H₂O₂/O₂ = 0.6x10⁻³ and HO₂/O₂ = 1.9x10⁻³. These were compared with the abundance ratios measured in the ρ Oph A dense core²⁴, where O₂ has also been detected and is likely to originate from radiolysis: H₂O₂/O₂ \approx HO₂/O₂ \approx 0.6x10⁻³. The H₂O₂ abundance relative to O₂ is clearly spot on but the value for the HO₂/O₂ ratio is 3x larger in the coma of 67P. We argue that this ratio should be either 0.6x10⁻³ (i.e., HO₂ has survived intact in the ice), or lower; it should not be larger as HO₂ cannot be formed to any significant degree through gas phase collisions of O₂ and H in the tenuous coma. This discrepancy is significant and, together with the absence of O₃, constrains the primordial radiolysis conjecture.

We believe that Bieler et al. would have had a hard time ruling out the ER mechanism. With the exception of the abundance issue, the dynamic production of O₂ from water ions is certainly more consistent with all other trends and can explain more observations than the primordial origin hypothesis. Furthermore, the ER mechanism produces energetic O₂⁻ negative ions, which reach Rosetta and enter the DFMS at hyperthermal energies between 10-50 eV. These ions collide with gold-coated surfaces inside the DFMS and can charge exchange with the surface to emerge as O₂⁺. This “Hyperthermal Surface Ionization” mechanism is well known in the surface dynamics community and there is even a commercial system sold. That is, O₂⁺ ions are produced inside the DFMS ionizer by a mechanism other than electron impact. If these ions are detected, they can lead to overestimation of O₂ abundance. We have verified that the DFMS has not been calibrated for this eventuality. We have revised the original text to point that out (p. 9):

However, we note that a unique feature of the ER reaction mechanism is its ability to produce energetic O₂⁻ ions, moving away from the nucleus towards the orbiting Rosetta spacecraft with kinetic energy between 10 and 50 eV. The double focus mass spectrometer (DFMS) entrance slit plate is biased in gas mode to reject ambient ions³². Positive bias will attract and accelerate O₂⁻ into the ionizer box. Depending on cometocentric distance, some or all O₂⁻ in transit to Rosetta will undergo photodetachment¹⁸, producing neutral O₂ molecules, which retain their kinetic energy and will enter the ionizer regardless of bias. Energetic collisions of the hyperthermal O₂⁻ ions or O₂ molecules with the gold-coated internal surfaces of the DFMS will produce O₂⁺ ions by surface re-ionization³³ (see Suppl. Fig. S11). It is worth pondering how O₂⁺ formed inside the DFMS by a mechanism other than electron impact ionization will contribute to the detected O₂⁺ signal³⁴.

Remark 2: Since the discovery of high amount of O₂, other scenarios have been explored, especially chemical routes. They should be discussed in the paper : Dulieu et al 2017, Taquet et al 2016.

ANSWER: These two papers, and a third one by Mousis et al., are now cited and discussed in the revision on page 8-9:

There have been other attempts at justifying the primordial O₂ formation and survival for 4.6 Billion years. Mousis et al.²⁷ considered the radiolysis of icy grains in the low-density proto-solar nebula (PSN), which may produce large amounts of O₂ though “its incorporation as crystalline ice is highly implausible”. These authors identified two extreme scenarios for dense and early PSN, which require galactic cosmic ray flux larger than possible over the lifetime of the PSN and O₂ trapping in clathrates. Taquet et al.²⁸ used sophisticated astrochemical models to compare various primordial O₂ formation mechanisms, proposing oxygen atom recombination at the surface of interstellar ices as a possibility, albeit under “warmer and denser conditions than usually expected in dark clouds”. Finally, Dulieu et al.²⁹ proposed that O₂ forms in situ during the evaporation of water ice via a dismutation reaction of co-evaporating H₂O₂. This mechanism requires the incorporation of primordial H₂O₂ in large amounts into the nucleus and its complete conversion into O₂ to be consistent with the low levels of H₂O₂ in the coma. All these mechanisms appear to be in conflict with the relative abundances of the related species O₃, H₂O₂, and HO₂.

Oxidized Fe, Si, and Pt surfaces are used to mimic olivine or pyroxene silicates or non volatile (organic) refractory materials. Actually, the ER mechanism seems to be transportable from SiO_x, FeO_x to Pt(O), but the generalization to cometary surfaces is unclear to me.

Question : Is it reasonable to think that any oxidized surface materials (quartz) would give similar results ?

Remark 3: The choice of these surfaces and their astrophysical relevance has to be discussed, and I wonder why silicates was not used for example.

ANSWER: We have revised the Methods section to explain the choice of analogs:

Thin oxides are used as analogs of cometary materials because beam experiments at high ion flux create localized surface charging that influences measurements of exit energies. A thin silicon oxide (~2-3 nm) allows removal of the surface charge by electron tunneling to a grounded substrate. This situation is not encountered in cometary materials (silicates, quartz, etc.) because of the presence of solar wind, which provides electrons to the surface to neutralize this charge. Furthermore, the oxygen atoms on the silicon oxide surface are bonded to Si in a similar fashion to surface oxygen atoms in silicates. We have verified the charging effect by scattering on titanium oxide, which is a semiconductor and has mild conductivity, but exhibits the same scattering behavior as silicon oxide (see Suppl. Fig. 12). On the other hand, elemental Fe has been found on 67P, and given the oxidizing conditions in the coma, oxidized Fe surfaces are to be expected.

Experimental methods and section: As far as I understood, the authors demonstrate that H₂O⁺ bombardment of oxidized Fe surfaces Si surfaces and Pt(O) produce HO₂⁻ which evolve into O₂⁻. They discuss about the possible H₂O-O* intermediate.

It is absolutely unclear that they have ever detected any O₂ molecules or, that, taken into account that ions are usually more easily detected, that there is an indirect method to conclude that O₂ is directly formed. On the contrary, it is clear that O₂⁻ have high kinetic energy that can be measured. My understanding is that mostly O₂⁻ is produced, and not O₂.

Remark 5: Please clarify this point. It is not correct to let the reader think that the paper is about O₂ if O₂⁻ is the real product and object of study.

ANSWER: Yes, indeed, only ions (positive and negative) can be detected in our experiments. Neutral O₂ is expected and usually forms in larger abundance than the ions. However, in order to detect neutrals, they must be ionized first—usually done by electron impact. Unfortunately, when neutral O₂ molecules move with high velocities (corresponding to 10-50 eV), they are extremely

difficult to ionize by electron impact due to abysmally low cross-sections. This is also happening inside the DFMS aboard Rosetta, this instrument cannot detect fast neutrals.

So, how do we know that neutral O₂ molecules are there? Because of electron detachment processes. If negative ions can form, they can lose electrons by auto-detachment due to high internal energy from the hard collision. If they lose one electron, they become neutral and cannot be detected. But if they lose two electrons, they become positively charged (O₂⁺) and can be detected. In fact, we do detect O₂⁺ in H₂O/SiO_x collisions, see Suppl. Fig.1i. The kinematics of O₂⁻ and O₂⁺ are similar because spontaneous electron loss does not cause inelasticity, which further confirms that one electron loss to form neutral O₂ is energetically feasible.

We have revised the abstract and the text to point out that what we detect is only O₂⁻. We also provide new references to electron photodetachment papers for comets (Chaizy et al.) and earth's atmosphere (Wisemberg et al.), which discuss the efficient conversion of negative ions to neutrals.

Abundant molecular oxygen was discovered in the coma of comet 67P/Churyumov-Gerasimenko. Its origin was ascribed to primordial gaseous O₂ incorporated into the nucleus during the comet's formation. This thesis was put forward after discounting several O₂ production mechanisms in comets, including photolysis and radiolysis of water, solar wind-surface interactions, and gas-phase collisions. Here we provide experimental evidence for an original Eley-Rideal reaction, which permits instantaneous O₂ formation in single collisions of energetic water ions with oxidized cometary surface analogs. The reaction proceeds by H₂O⁺ abstracting a surface O-atom, forming an excited precursor state, which dissociates to produce O₂⁻. Subsequent photo-detachment leads to O₂, whose presence in the coma may thus be linked directly to water ions and their interaction with the solar wind. This abiotic O₂ production mechanism is consistent with reported trends in the 67P coma and raises awareness of the role of energetic negative ions in comets.

Remark 4: Because O₂⁻ is a key tracer of the ER mechanism, a careful attention should be paid to the presence of this anion in cometary atmospheres. To my knowledge, no O₂⁻ detection have been reported, whereas others anions (H-, OH-, CN-) have been reported (ref 21). It is certainly a point to discuss, especially if no O₂⁻ have been detected.

ANSWER: We wholeheartedly agree! With the exception of H-, **no other negative ions** have been reported for comet 67P. If the referee has knowledge of report(s) disclosing OH- and CN- in the 67P coma, will she/he please tell us **where** these finding have been published? The referee quotes ref. 21 of our original submission, which refers to: "21. Balsiger, H. *et al.*, *Space Sci. Rev.* **128**, 745-801 (2007)" but this paper is from 2007, before Rosetta's arrival to the comet. Fuselier et al. (revision, ref.31) reports only **positive** ions found on 67P and, surprisingly, excludes O₂⁺. To the best of our knowledge, data on negative ions have yet to appear.

In the absence of any publication from the Rosetta mission on negative ions (except H-), we have to rely on measurements on 1P/Halley, whose coma has been visited by Giotto. We have revised the Discussion in the manuscript to state on page 7:

Negative ion production. We have uncovered high-energy reaction channels for dynamic production of negative ions from collisions of energetic water ions with oxidized surfaces. The latter surfaces include: SiO_x, FeO_y, Pt(O), NiO_z, Pd(O), Au(O), TiO_w (see also Suppl. Figs. S10, S11, S12)]. Such interactions are applicable to plasmas and astrophysical environments, whenever H₂O⁺ ions are encountered with kinetic energies between 50 and 300 eV. We propose that the scattering interactions occur in cometary comae during periods of activity, where they produce energetic negative ions, including: O⁻, OH⁻, O₂⁻, and HO₂⁻. The latter two ions, in particular, are produced by a novel ER reaction, and contribute to the O₂ abundance

in the coma after photodetachment^{18,19}. The lifetime of O_2^- against photo-detachment is 2.6 s (at 1 AU)¹⁹, which suggests that O_2^- should be able to reach Rosetta. Negative ions should be present in the coma of 67P but they have yet to be reported, barring H^- (ref. 20). Thus, our work actually *predicts* the existence of O_2^- , and HO_2^- in the coma at distances sufficiently close to the nucleus to prevent photodetachment. Negative ions of cometary origin have been detected in the coma of comet 1P/Halley, though without sufficient mass-resolution to distinguish individual ions¹⁸. Three broad peaks were observed, which were denoted as the 17-, 30-, and 100-AMU peaks. Chaizy et al.¹⁸ argued that the first peak included O^- and OH^- , while the second peak comprised CN^- . Furthermore, these authors considered several negative ion production mechanisms and found them inadequate to explain the signal intensity observed in the Halley coma. We propose that energetic water ion scattering on the nucleus surface or dust grains in the coma of 1P/Halley could populate both the 17- and 30-AMU peaks, and that the latter peak must have also included O_2^- and HO_2^- .

The referee's request creates an interesting predicament: we are effectively predicting that O_2^- and HO_2^- should exist in the 67P coma, and if they do, that would constitute a validation of the ER reaction and our claims. We accept the challenge...

Line 161-171 (concluding paragraph): I don't really understand what do the authors mean. It is very loose. "some or all O_2^- may be neutralized by photodetachment" contains no information but is central, if orders of magnitude are provided. O_2^- has a relatively low electron affinity, therefore, its photodetachment rate should be high under solar light (IR and Visible). It is thus reasonable to think that 50eV neutral O_2 will rapidly escape from the atmosphere or strongly interact with it. All these aspects should be discussed more carefully in order to make the link between the experiments and the observations.

ANSWER: We have revised the Discussion in the manuscript in two places to clarify our position on photodetachment of O_2^- . We state on page 7:

The latter two ions, in particular, are produced by a novel ER reaction, and contribute to the O_2 abundance in the coma after photodetachment^{18,19}. The lifetime of O_2^- against photo-detachment is 2.6 s (at 1 AU)¹⁹, which suggests that O_2^- should be able to reach Rosetta.

Furthermore, we have revised the last paragraph of the Discussion section to point out how negative ions interact with the internal surfaces of the DFMS instrument:

However, we note that a unique feature of the ER reaction mechanism is its ability to produce energetic O_2^- ions, moving away from the nucleus towards the orbiting Rosetta spacecraft with kinetic energy between 10 and 50 eV. The double focus mass spectrometer (DFMS) entrance slit plate is biased in gas mode to reject ambient ions³². Positive bias will attract and accelerate O_2^- into the ionizer box. Depending on cometocentric distance, some or all O_2^- in transit to Rosetta will undergo photodetachment¹⁸, producing neutral O_2 molecules, which retain their kinetic energy and will enter the ionizer regardless of bias. Energetic collisions of the hyperthermal O_2^- ions or O_2 molecules with the gold-coated internal surfaces of the DFMS will produce O_2^+ ions by surface re-ionization³³ (see Suppl. Fig. S11). It is worth pondering how O_2^+ formed inside the DFMS by a mechanism other than electron impact ionization will contribute to the detected O_2^+ signal³⁴

General conclusion:

The authors performed a valuable new piece of science, and they show how H_2O^+ (and similar impactors) in the energy range of 100eV can produce HO_2^- which evolves into O_2^- of few tens of kinetic energy for the larger part of the incident energies, after abstraction of an oxygen atom of

oxidized surfaces of Fe Si and Pt(O). They clearly show that it is not simple sputtering but really the ER mechanism. However, even if “all necessary conditions for such reactions are met are present in the cometary atmosphere” (and we can discuss the point of the surfaces), I found that in there is no key to understand how much this process can explain the in-situ measurement of Bieler et al. Moreover, I found the gap between the title and the experiments which support the real scientific discussion very deep. The contrast between the high level of the experimental discussion and the approximative astrophysical arguments is obvious.

Yes, it is reasonable to think that the astrophysical scenario proposed here may contribute to the production of O₂. Is it the real conclusion of the authors ? Actually I don't know, I don't understand what is their final conclusion.

We have added a Conclusion section (page 10) to spell out several important points of our paper:

Conclusions

Energetic water ions in cometary comae, produced and accelerated by solar wind, can drive scattering interactions on cometary surfaces that alter the relative speciation in the coma. Collision-induced water dissociation can increase negative ion densities (O⁻, OH⁻). Abstraction of chemisorbed oxygen from oxidized surfaces by water ions can also produce dynamically O₂⁻ and HO₂⁻, by means of a previously unknown Eley-Rideal reaction. Kinematic analysis of the reaction products provides indirect evidence for the elusive oxywater state as a reaction intermediate, which may form during the hard collision, then dissociate promptly on the rebound from the surface. When this reaction occurs in comets, it can populate the coma with energetic O₂⁻, which is converted to molecular O₂ by photo-detachment. This new abiotic way to produce molecular O₂ informs our understanding of cometary chemistry and could be important in other astrophysical environments.

At this stage I suggest the authors to choose whether their claim is an ambitious astrophysical scenario and therefore to convince with adequate order of magnitudes that this scenario is solid, or to focus on experiments, as they actually mostly did, but thus to better show where is the real novelty. I also suggest to be clear about the role of anions.

ANSWER: We deeply appreciate this suggestion and all thoughtful comments above. It is seldom that a reviewer provides this much feedback. We have revised our manuscript extensively and we believe that the focus of the work has become clearer. This is a case of a novel reaction impacting cometary chemistry. Both aspects of the work must come out clear and withstand the scrutiny of two disparate and tough communities. The double challenge is indeed very tall.

Question related to remark 1: Please provide explicitly the H₂O⁺/H₂O ratio in cometary atmosphere (or a range because it is space and time dependent). Give an order of magnitude of the total efficiency of the reaction (How many O₂ per H₂O⁺ sent to the surface).

ANSWER: The measuring efficiency of the reaction requires careful calibration, not possible at this time. A major difficulty relates to integrating the scattering signal (O₂⁻) over 3-dimensions, which requires experimental capabilities not available on our fixed geometry scattering apparatus.

Suggestion : What about the proposed mechanism and the elusive detection of O₂ in the ISM ?

ANSWER: Gaseous O₂ has been detected in two interstellar clouds so far (the Orion nebula and the ρ Ophiuchi A core) and is generally known to have surprisingly low abundances. See work of Goldsmith et al. 2011, Yildiz et al. 2013, Bergman et al. 2011, Parise et al. 2012.

We believe that the origin of O₂ in interstellar clouds is radiolysis of water. We see no mechanism for ionizing and accelerating water ions to energies required for ER reactions in interstellar clouds. Until the Rosetta mission, it was unknown that such energetic ions even existed in comets.

Reviewer #3 (Remarks to the Author):

It has heretofore not been possible to explain the large amount of O₂ found in the coma of comet 67P using standard theories and mechanisms. The authors provide experimental evidence for the production of O₂ via a previously unexplored Eley-Rideal mechanism involving cationic water collisions with oxide surfaces. Gas-surface scattering experiments are performed in UHV involving high-energy ion beams with the detection of scattered ions via mass spec.

H₂O⁺ is scattered from amorphous Si and Fe oxide surfaces over a wide range of energies, and shown to produce O₂⁻ and HO₂⁻. The authors make a reasonable argument against the formation of O₂⁻ via sputtering. Studies of H₂O⁺ scattering from O-covered Pt surfaces, using different isotopic combinations for the O atoms, are consistent with this argument. The authors propose that the incident H₂O⁺ abstracts a chemisorbed O atom from the oxide surface (an Eley-Rideal reaction) to form oxywater, H₂O₂. The oxywater in turn rapidly dissociates into H⁺ and HO₂⁻, and the HO₂⁻ fragment eventually dissociates into H and O₂⁻. Unfortunately, there is no direct evidence for the formation of the H₂O₂. However, its existence has been argued using high-level ab initio theory, and it is reasonable that any H₂O₂ formed via this pathway would be very highly excited, and would likely break into more stable products. Both H⁺ and HO₂⁻ are detected in the experiments, and in the comet. The most compelling evidence comes from the kinematics of the scattered O₂⁻ and H⁺. The energies of these scattered products are consistent with the proposed Eley-Rideal mechanism, using standard binary collision theory to compute the scattering energy of the H₂O₂ parent.

I cannot comment on the astrophysical or experimental merits of this paper, but the proposed Eley-Rideal mechanism appears to be plausible, new, interesting, and consistent with the available data. Obviously much work needs to be done, but I think this paper is suitable for publication.

ANSWER: We appreciate the recommendation and thorough reading of the manuscript. This is a case of a novel reaction impacting cometary chemistry. Both aspects of the work must come out clear and withstand the scrutiny of two disparate communities. The double challenge is indeed very tall. We are glad to obtain confirmation that we pass the scrutiny of the surface dynamics community. Thank you.

REVIEWERS' COMMENTS:

Reviewer #1 (Remarks to the Author):

The revised version of the paper is improved a lot, considering also the comments from the other Reviewers.

The authors have addressed all the questions in a satisfactory way.

I suggest the paper for publication in Nature Communications.

Reviewer #2 (Remarks to the Author):

The authors have significantly changed their main message and especially clarified and emphasised the possible key rôle of anions in comets atmospheres, which is a very relevant point. It is not only because I am pleased that authors followed my suggestion, I also hope that they consider now that their work is more accurate in the implications, and as such harder to refute. My second concern was about the absence of estimate of total efficiency of their proposed mechanism. I am a bit disappointed to have no clues to estimate it. I guess that authors could have given boundary limits. However I understand their experimental difficulty. But still I would prefer them to give a rough estimate (even wrong), more than seeing this mechanism included 'quick and dirty' in a forthcoming cometary model. Although I disagree, I respect their point of view.

I have still minor points of disagreement/question that authors may wish to think about.

- Please remove the word 'instantaneous' from the abstract. I understand why it is there, but it creates confusion.

- I183: O₂⁻ can reach Rosetta. Actually, photodetachment is one of the decay channel. I don't know about collisions rate, but I know the fragility of negative ions. Absence of detection of O₂⁻ in Rosetta would not be necessary a disproof of your suggested O₂ formation mechanism. It is also why I did not consider your remark on DFMS as a strong argument, in your previous version.

- I would be more cautious about the significance of HO₂/H₂O₂ ratios. Formation pathways are actually important but, destruction pathways are also crucial. The absence of O₃ is a strong indication, but taking into account the difference in mass spectroscopy detection and radio spectroscopy, I would not consider H₂O₂ and HO₂ ratios as strong indications. Factor of 3 may be significant, but I would not bet too much on it. Also note that HO₂ is steadily formed but O₂+H reaction on grains.

RESPONSE TO REVIEWERS

Reviewer #2 (Remarks to the Author):

The authors have significantly changed their main message and especially clarified and emphasised the possible key rôle of anions in comets atmospheres, which is a very relevant point. It is not only because I am pleased that authors followed my suggestion, I also hope that they consider now that their work is more accurate in the implications, and as such harder to refute.

ANSWER: We are deeply thankful for the valuable comments and suggestions. Indeed, the revised manuscript has more clarity and focus.

My second concern was about the absence of estimate of total efficiency of their proposed mechanism. I am a bit disappointed to have no clues to estimate it. I guess that authors could have given boundary limits. However I understand their experimental difficulty. But still I would prefer them to give a rough estimate (even wrong), more than seeing this mechanism included 'quick and dirty' in a forthcoming cometary model. Although I disagree, I respect their point of view.

ANSWER: Any estimate would be premature without knowledge of the flux and energy distributions of all water-derived ions (O^+ , HO^+ , H_2O^+ , H_3O^+). Point-in-case: newly cite Ref.37 presents cometary measurements of the H_3O^+/H_2O^+ density ratio, which varies from under one to almost 100. H_3O^+ can also participate in Eley-Rideal reactions similar to H_2O^+ to produce molecular oxygen. Preliminary experiments suggest that the reaction efficiency for H_3O^+ is the same as that for H_2O^+ , which means that H_3O^+ may actually be a significant contributor to cometary O_2 .

I have still minor points of disagreement/question that authors may wish to think about.
- Please remove the word 'instantaneous' from the abstract. I understand why it is there, but it creates confusion.

ANSWER: The word “instantaneous” has been removed from the abstract.

- 1183: O_2^- can reach Rosetta. Actually, photodetachment is one of the decay channel. I don't know about collisions rate, but I know the fragility of negative ions. Absence of detection of O_2^- in Rosetta would not be necessary a disproof of your suggested O_2 formation mechanism. It is also why I did not consider your remark on DFMS as a strong argument, in your previous version.

ANSWER: We appreciate the viewpoint. Depending on the cometocentric distance, some or all of O_2^- will be neutralized. That is why our argument includes neutralized O_2 entering the DFMS at hyperthermal energy. Hyperthermal surface ionization of O_2 will produce O_2^+ inside the ionizer. If this O_2^+ signal is detected, it must be distinguished from O_2^+ produced by electron impact, or it will lead to significant overestimation of the O_2 density.

- I would be more cautious about the significance of HO₂/H₂O₂ ratios. Formation pathways are actually important but, destruction pathways are also crucial. The absence of O₃ is a strong indication, but taking into account the difference in mass spectroscopy detection and radio spectroscopy, I would not consider H₂O₂ and HO₂ ratios as strong indications. Factor of 3 may be significant, but I would not bet too much on it. Also note that HO₂ is steadily formed but O₂+H reaction on grains.

ANSWER: We appreciate the viewpoint. We toned down the discussion to take a more cautious approach to the interpretation of the HO₂/H₂O₂ ratios. We changed the text to point out that cometary O₂ can itself drive the production of HO₂ by a hydrogen abstraction reaction on the cometary surface and thus contribute to the increase of the HO₂/H₂O₂ ratio:

“This difference suggests that HO₂ is formed at higher rates than it can be destroyed, thus accumulating in the coma. It is likely that HO₂ forms by a mechanism different from or perhaps in addition to that operating under interstellar conditions. Apart from the mechanism discussed in this communication, there is actually another reaction mechanism enabled by the presence of O₂ in the coma, which involves a different Eley-Rideal reaction³⁵. Like H₂O⁺, photo-ionized O₂⁺ can be picked up by solar wind and accelerated back to the comet, where it can abstract atomic H from cometary materials to form HO₂.”